# PAL-UI: Planning with Active Look-back for Vision-Based GUI Agents

## Abstract

Graphical User Interface (GUI) agents powered by Multimodal Large Language Models (MLLMs) promise human-like interaction with software applications, yet long-horizon tasks remain challenging due to memory limitations. Existing approaches either truncate history or rely on simple textual summaries, which risk losing critical information when past visual details become necessary for future decisions. In this paper, we propose **PAL-UI** (**P**lanning with **A**ctive **L**ook-back), a novel framework that enables GUI agents to adaptively retrieve past observations when required. PAL-UI combines a dual-level summarization agent, capturing both observation-level cues and action-level outcomes, with a dedicated retrieval tool that allows the agent to recall specific historical screenshots during planning. We curate a step-level instruction dataset of 8.6K samples from mobile GUI navigation trajectories and train **PAL-UI-3B** and **PAL-UI-7B** models based on Qwen2.5-VL. Extensive experiments demonstrate that PAL-UI significantly outperforms baseline models and prior methods in mobile GUI navigation tasks, even under data-efficient settings. Moreover, PAL-UI exhibits strong cross-domain generalization, achieving notable improvements in web navigation without additional training. Our work highlights the potential of active memory retrieval for long-horizon planning capabilities of vision-based GUI agents.

## 1 Introduction

Large Language Models (LLMs) have dramatically advanced the capabilities of AI systems in recent years (Brown et al., 2020; Zhao et al., 2023). This progress has spurred the development of GUI agents, i.e., autonomous agents that perform tasks via graphical user interfaces (GUI) (Wang et al., 2024; Hong et al., 2024). Early paradigms for LLM-driven GUI agents typically relied on converting visual interface information into textual form (e.g., reading an app's accessibility tree or metadata) so that a language model could process it (Nakano et al., 2021; Zhang & Zhang, 2023). However, such text-based representations often require external modules and inject a large number of additional tokens into the context, limiting efficiency and fidelity. With the emergence of Multimodal LLMs (MLLMs) that can directly handle images as input (Liu et al., 2023), a new vision-based GUI agent paradigm has surfaced (Gou et al., 2024; Xu et al., 2024). These agents perceive raw screenshots of the interface and simulate human-like operations on the GUI, enabling end-to-end interaction without intermediate text conversions.

A central challenge for long-horizon GUI tasks is how to incorporate memory of past observations and actions into the agent's planning. In traditional text-based agents, a common strategy is to append the entire interaction history to the input context for planning (Yao et al., 2023). Unfortunately, directly extending this to vision-based agents is infeasible. Visual observations are much heavier than text, i.e., each image input to an MLLM is encoded into a large number of tokens (Bai et al., 2025), quickly exhausting the model's context length. Moreover, current MLLMs struggle to reason effectively over many images at once; recent studies show that their capability to handle multiple simultaneous images remains quite limited (Zhao et al., 2024). Simply providing all past screenshots as input can overwhelm the model with redundant information and obscure the relevant details. As a result, existing vision-enabled GUI agents resort to very restrictive memory usage: some only feed the most recent action or a brief textual summary of it into the context (Chen et al., 2024; Xu et al., 2024), while others retain only a few of the last screenshots as visual memory (Qin et al., 2025).

These ad-hoc strategies risk losing critical information from earlier steps and ultimately constrain the agent's performance on complex, multi-step tasks.

To address the above limitations, we propose a new paradigm called **P**lanning with **A**ctive **L**ookback (**PAL-UI**). The key idea is to empower the GUI agent to actively retrieve and consult pertinent details from its history when needed, rather than carrying the full burden of the past at every step. Concretely, we equip the agent with a tool interface that can fetch a specific historical observation (a screenshot from a previous step) on demand during the planning process. The agent operates in an iterative loop: it maintains a compressed memory of the task so far (i.e., a concise textual summary of past interactions), which provides a lightweight context for the language model. While this summary covers the high-level history, the agent can look back using the tool to retrieve detailed visual information from any prior step it deems important for the current decision. This active look-back mechanism mimics how a human might momentarily glance at a past screen or recall a specific detail when uncertain about the next action. By integrating tool-based retrieval into the planning loop, our agent can leverage extensive historical information when necessary, without suffering from the context explosion or distraction issues of naively large visual contexts.

Training an agent to perform active look-back introduces unique challenges, as standard demonstration data lack both tool-use annotations and explicit reasoning traces. To bridge this gap, we construct a synthetic instruction-tuning dataset that augments raw trajectories from AndroidControl (Li et al., 2024) with tool-calling behavior. Concretely, we propose a four-stage *deliberated look-back framework* that guides a stronger teacher model to simulate when and how an agent should retrieve past observations. From these curated trajectories, we filter for correctness, rebalance samples to prevent retrieval recency bias, supplement with non-retrieval cases to avoid overfitting, and finally standardize into a structured dialogue format. The resulting dataset comprises **8.6K** high-quality, step-level trajectories. we fine-tune a Qwen2.5-VL (Bai et al., 2025) backbone on this dataset, obtaining our PAL-UI agents in two sizes (**PAL-UI-3B** and **PAL-UI-7B**), both of which acquire the ability to reflect, decide when to look back, and act effectively in long-horizon GUI tasks.

We evaluate our PAL-UI agent on a broad set of GUI navigation benchmarks. Experimental results demonstrate that PAL-UI significantly outperforms the base MLLM (without active look-back) on long-horizon mobile UI tasks, achieving new state-of-the-art performance under comparable training data settings. Notably, by effectively leveraging historical context, our method yields higher success rates than prior methods even with far fewer training examples. Moreover, although our training data and design focused on mobile app environments, the PAL-UI agent exhibits strong zero-shot transfer to other domains, such as web browser interfaces. It substantially improves task success on web-based GUI benchmarks compared to baselines, highlighting the generality of our approach.

## 2 RELATED WORK

### 2.1 GUI AGENTS

With the success of large language models (LLM) and multimodal large language models (MLLM), GUI Agents (Gou et al., 2024; Qin et al., 2025) have achieved significant advancement across various GUI platforms. Early GUI Agents (Zhang & Zhang, 2023; Zheng et al., 2024a) usually rely on structured information such as HTML or accessibility trees for element localization, making them difficult to generalize across different platforms. Consequently, researches shift toward vision-based GUI Agents (Gou et al., 2024; Xu et al., 2024), which simply take screenshots as observations and interact with interfaces through human-like mouse and keyboard actions. Such paradigms enable end-to-end automation for cross-platform tasks, making progress toward broader applicability.

### 2.2 MEMORY MANAGEMENT FOR GUI AGENTS

Previous LLM-based agent (Yao et al., 2023) systems usually manage memory simply by appending history information (observations and actions) directly to the input context. However, such paradigm introduces several technical challenges for vision-based GUI Agents. First, the observations for GUI agents exist in the form of screenshot images (Cheng et al., 2024). Storing all history screenshots can lead to high token costs as images consume a substantial number of tokens (Bai et al., 2025). Second, such approach will introduce multi-image inputs for the agent model, potentially impairing the model's reasoning capability (Zhao et al., 2024). Consequently, most existing approaches (Chen

et al., 2024; Xu et al., 2024) retain only past actions or the few most recent past observations as memory (Qin et al., 2025). To address the limitation, we propose PAL-UI, a paradigm that allows agents to plan with active look-back. In this way, the agent can actively retrieve detailed information from history during inference, thereby mitigating the potential information loss in previous paradigms.

## 2.3 MLLM-based Tool-use Agents

Enhancing MLLMs through tool calling has recently emerged as a popular direction, as external tools enable MLLMs to transcend their capability bottlenecks and improve their performance. Early approaches (Wu et al., 2023; Yang et al., 2023c) often employed training-free prompting methods to invoke tools and enhance the model's visual perception abilities. Subsequent work, such as LLaVA-Plus Liu et al. (2024) and GPT4tools (Yang et al., 2023b), further strengthened MLLMs' tool-use capabilities by synthesizing high-quality tool-calling trajectories and performing supervised fine-tuning. With the development of slow-thinking reasoning paradigms (Jaech et al., 2024; Guo et al., 2025), more recent studies have shifted toward a "Thinking with images" approach (Su et al., 2025; Zhou et al., 2025), where tools are invoked during the reasoning process to edit input images, leading to significant improvements in the model's reasoning performance. As for MLLM-based GUI agents, early methods (Zheng et al., 2024a) often relied on Set-of-Marks (SoM) (Yang et al., 2023a) or accessibility trees to provide additional on-screen information. Recently, FOCUS (Tang et al., 2025) proposed a dual-system framework comprising fast and slow prediction mechanisms to enhance GUI grounding. To the best of our knowledge, we are the first to leverage external tools to improve long-horizon planning ability in GUI agents through an active look-back mechanism.

## 3 Problem Formulation

We consider a GUI-based sequential decision process where an agent must achieve a specified goal by interacting with a user interface. Formally, at each time step $i$, the agent receives an observation $o_i$ of the current GUI state (e.g., a screenshot or UI view) and selects an action $a_i$ from the set of possible interface actions (such as clicking a button, entering text, or scrolling). Executing action $a_i$ changes the interface state, leading to a new observation $o_{i+1}$. The process continues until the agent accomplishes the global goal $G$, which is given as part of the task (typically a high-level instruction or target outcome), or until a maximum number of steps is reached.

Each task thus forms a trajectory $\tau = (o_0, a_1, o_1, a_2, o_2, \ldots, a_T, o_T)$, where $o_0$ is the initial observation and $T$ is the total number of steps. The core challenge in this setting lies in *long-horizon planning and memory management*: as $T$ grows, the agent accumulates a large history of observations (each potentially high-dimensional, such as images with text) and actions. A naive strategy of feeding the entire raw history into a vision-based agent quickly becomes impractical due to context length limits and redundant information. In fact, without a mechanism to compress and retrieve relevant information from past observations, an LLM-based planner may suffer performance declines when important details from earlier steps get lost in a sea of tokens. Our goal is to enable the agent to retain critical information from its interaction history and actively look back to past states when necessary, all while staying within feasible context lengths.

We follow previous studies (Bai et al., 2025; Qin et al., 2025) to adopt a unified action space for interacting with the GUI interface. Specifically, we introduce the following three types of actions:

- General actions across all platforms: `Click`, `Type`, `Scroll`, `Drag`, `Wait`, and `Finished`;

- Actions for mobile platform: `LongPress`, `OpenApp`, `PressHome`, and `PressBack`;

- Actions for web platform: `Hotkey`, `LeftDouble`, and `RightSingle`.

Note that some actions may require an action-related content (e.g., coordinates for `Click` and textual string for `Type`). During inference, we prompt the GUI agent to generate a reasoning process and a predicted action. Following Lu et al. (2025), the reasoning process and predicted action are encircled in `<think></think>` and `<tool_use></tool_use>` tokens, respectively.

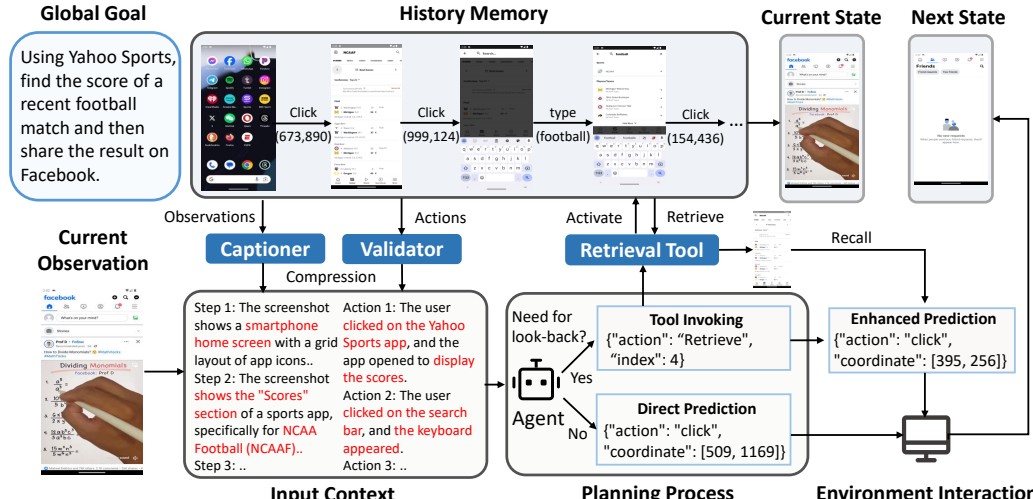

Figure 1: The illustration of our proposed PAL-UI agent. We utilize an observation-level captioner and an action-level validator for memory compression. Then, equipped with a retrieval tool, the agent is able to actively recall detailed visual information from past memory at inference time.

## 4 APPROACH

We propose **PAL-UI** (**P**lanning with **A**ctive **L**ook-back), a framework to enhance the long-horizon planning capabilities of a GUI agent by combining dual-level summarization of the history with an active retrieval mechanism. Figure 1 gives an overview of our approach. In particular, PAL-UI compresses the interaction history into a succinct textual memory and equips the agent with a special tool to fetch detailed visual information from past steps on demand. The agent is trained via supervised fine-tuning on a dataset of expert trajectories augmented with tool-use demonstrations. We organize our approach as follows: first, we introduce the construction of the summarization agent and the retrieval tool (Section 4.1); second, we explain how the agent plans with active look-back at inference time (Section 4.2); finally, we describe the training data generation pipeline used to impart these capabilities to the agent (Section 4.3).

### 4.1 TOOL CONSTRUCTION AND SUMMARY AGENT

To prevent the agent's context from being overwhelmed by redundant visual tokens, we compress the history at two levels, i.e., observations and actions, using a **dual-level summary agent**. Meanwhile, we introduce a **memory retrieval tool** that the agent can invoke to recover raw observations from any past step when detailed information is needed.

**Observation-level Captioner.** We first summarize each new visual observation $o_i$ to capture the key information relevant to the task. Even though a GUI screenshot may contain crucial cues for the long-term goal, these cues are often hidden among numerous UI elements. Simply appending every raw image $\{o_1, o_2, \ldots, o_{i-1}\}$ to the prompt would rapidly exhaust the context window as $i$ grows, and the model could struggle to locate important details in the noise. Therefore, for each step we generate a concise observation caption that highlights salient information in $o_i$ (especially text on the screen or unique interface changes) in relation to the global goal $G$. We employ the large vision-language model Qwen-2.5-VL (Bai et al., 2025) as our captioning module, due to its strong OCR capability and instruction-following performance. The captioner is prompted with the current screenshot $o_i$ and the task goal $G$ to produce a brief description focusing on the most relevant UI content (e.g., widget labels, displayed messages, or enabled/disabled states).

**Action-level Validator.** In addition to summarizing observations, we also summarize the outcome of each action to form an action memory. Prior works on GUI agents often record only the action sequence as history (Chen et al., 2024; Xu et al., 2024), but without the surrounding state context,

a plain list of actions provides limited insight into the task progress. Moreover, those approaches typically do not verify if an action succeeded, leaving the agent prone to repeating failed actions or getting stuck in loops. To address this, we introduce an action-level validator, which produces a compact description of the agent's last action $a_i$ and evaluates its execution result by comparing the interface before and after the action. Using the same Qwen-2.5-VL model, we prompt it with the triplet: current step's action $a_i$, the pre-action observation $o_i$, and the post-action observation $o_{i+1}$. The validator then generates a brief assessment, for example, "The user typed 'football' into the search bar, and the search results for related content are displayed.". This summary serves two purposes: it clarifies the intent of $a_i$ in natural language (which can be easier for the LLM to reason with than raw action code) and it confirms whether $a_i$ achieved the expected effect on the GUI state.

**Memory Retrieval Tool.** While the dual-level summaries dramatically condense the history, converting rich visual observations into text can inevitably lose some details, e.g., the exact appearance of a screen or an unseen UI element that later becomes relevant. To allow the agent to recover such details when needed, we implement a Retrieve tool (`Retrieve`) for active look-back. The Retrieve tool can be invoked with a past step index $j$ and returns the observation $o_j$ (e.g., the screenshot image at step $j$ ($j < i$)) to the agent's context. We add this tool to the agent's action space, so it can decide at any point to retrieve a past screen instead of executing a normal GUI action. The retrieved image $o_j$ is then appended to the agent's current context, enabling the model to re-inspect that state before continuing the plan. In essence, the agent has the choice to temporarily step back and "refresh its memory" of a previous interface state.

## 4.2 ACTIVE LOOK-BACK PLANNING PROCESS

With the ability to summarize history and retrieve past observations, the agent plans actions in a think-act loop that actively looks back when necessary. At each step $i$, the agent's context includes: (1) the compressed memory $m_i$ up to step $i$ (consisting of relevant summaries of past observations and actions), (2) the current raw observation $o_i$ (the latest screenshot of the GUI), and (3) the task goal $G$. Using this context, the agent generates either a concrete GUI action $a_i$ or a retrieval query. There are two possible outcomes for the planning at step $i$:

• **Direct Action Prediction**: In the typical case, the agent uses the context $\{m_i, o_i, G\}$ to directly predict the next GUI action $a_i$ (discussed in Section 3). This prediction is informed by the distilled knowledge in the summaries such as what has been done so far and what key information is on the current screen, allowing the agent to decide the best next step toward the goal.

• **Active Retrieval then Action**: If the agent is uncertain or needs more detail from a previous state, it can choose to invoke the `Retrieve` action instead of immediately outputting a GUI action. For example, the model might decide it needs to "*look back at the login screen (step 2) to recall the exact error message*" before deciding how to proceed. When the retrieve tool is called, the specified past screenshot $o_j$ is fetched and added to the context. The planning then continues with the augmented context $\{m_i, o_i, o_j, G\}$, and the agent produces the final action $a_i$ based on both the current state and the newly recalled information from step $j$. By integrating this look-back step into the reasoning process, the agent can significantly improve its prediction accuracy for $a_i$, especially in situations where subtle details from earlier in the task are needed to choose the correct action.

Overall, this planning with active look-back mechanism enables the agent to dynamically balance *remembering* (through compact summaries) and *recalling* (through targeted retrieval) as it navigates toward the global task goal.

## 4.3 TRAINING DATA CONSTRUCTION

Equipping the agent with the above capabilities requires appropriate training data that demonstrates when and how to use the summarization context and retrieval tool. However, standard human demonstration data do not contain examples of tool usage or reasoning traces. We therefore create a synthesized fine-tuning dataset of tool-augmented trajectories via a combination of distillation from a stronger model and careful data curation. The process consists of four main steps:

**Seed Trajectory Collection.** We begin with a large set of human demonstration trajectories from the AndroidControl (Li et al., 2024) training dataset, each consisting of a sequence of GUI screen-

shots $\{o_0, o_1, ..., o_T\}$, human actions $\{a_1, ..., a_T\}$, and a global task goal $G$. These serve as the ground-truth action sequences that our agent should ideally replicate. For each step in each trajectory, we run our dual-level summary agent on the ground-truth trajectory to generate the compressed memory $m_i$ summarizing all history up to step $i$ (from 0 to $T$).

**Tool-Use Calling Curation.** Given that seed trajectories consist only of observations and ground-truth actions, they lack explicit tool-use steps. To construct training trajectories that include retrieval behavior, we require the teacher model to simulate when and how an agent would invoke the `Retrieve` tool. However, prompting the teacher model directly to produce tool calls proved ineffective, as the base model was not pretrained on such interactions. To address this gap, we design a four-stage *deliberated look-back framework* that gradually guides the teacher to integrate retrieval into its reasoning. At each step, the teacher model is prompted to perform the following stages:

• **History Revision**: The model reviews the compressed memory $m_i$ to assess task progress toward the global goal $G$. This ensures a clear understanding of what has been achieved so far.

• **Candidate Proposals**: The model proposes several plausible next actions based on the current observation $o_i$, encouraging it to enumerate possible strategies rather than committing prematurely.

• **Confidence Evaluation**: The model reflects on how confident it is in each proposed action and whether it feels uncertain enough to warrant checking something in the past. If the model determines that a specific detail from an earlier step is needed, it should invoke the `Retrieve` tool.

• **Tool-Use Action Prediction**: Depending on the previous stage, the model will receive the retrieved observation $o_j$ (from step $j$) if retrieval was requested and then outputs $a_i$ with the additional context.

**Data Filtering and Balancing.** After synthesizing a large set of trajectories with tool calling, we then filter and balance this data to focus on high-quality examples. First, we discard any sample where the teacher's final predicted action $a_i$ does not match the ground-truth human action for that step, ensuring our training data only contains correct predictions. Among the remaining samples, we identify those in which the teacher actually utilized the retrieval tool and succeeded in choosing the correct action afterward. We found about 4.3K such high-quality tool-use cases. Since the teacher model tended to prefer looking at very recent steps, which might bias the agent to only look back one step, we increased the sampling weight of samples where the retrieved step $j$ was further back in history. Finally, to prevent the agent from overusing the tool when it's not necessary, we also include 4.3K high-quality samples where the teacher solved the step without any retrieval (direct action with correct outcome). This yields a balanced dataset of roughly 8.6K step-level samples, half with tool use and half without, all with correct decisions.

**Compilation and Formatting.** After obtaining data samples, we compile them into the Qwen2.5-VL format for supervised fine-tuning (SFT) (Bai et al., 2025). At each step, the input consists of the system prompt, compressed memory $m_i$, and the current observation $o_i$, while the output is composed of the reasoning process and actions. To make a coherent reasoning, we first employ a strong LLM, Qwen3-32B (Yang et al., 2025), to synthesize all model responses from each stage of the above deliberated look-back framework into a logical natural language explanation. This explanation will serve as the thinking process, enclosed in `<think></think>` tokens. Finally, the action (`Retrieval` or other types of actions in Section 3) is encircled by `<tool_use></tool_use>` tokens and the retrieved screenshot will be injected into the next step. This standardized format directly teaches the agent how to reflect, when to look back, and how to act effectively.

## 5 EXPERIMENTS

### 5.1 EVALUATION BENCHMARKS

We evaluate our PAL-UI agent on two high-level GUI planning tasks: **AndroidControl-High** (Li et al., 2024) and **GUI-Odyssey** (Lu et al., 2024). For all tasks, we adopt a subtask-level evaluation paradigm where the model predicts the next action based on the global task goal, current screenshot, and historical memory. Following prior work (Gou et al., 2024; Xu et al., 2024), we report the *type match score (Type)*, *grounding accuracy (GR)*, and *step success rate (SR)* for the two benchmarks.

Table 1: Results on two mobile GUI datasets. "Method" indicates the training and inference method. "ZS", "SFT" and "RFT" are short for zero-shot, supervised fine-tuning, and reinforcement fine-tuning, respectively. **Bold** and underline denote the best and second best results.

| Model | Method | AndroidControl-High | | | GUI-Odyssey | | | Overall |
|---|---|---|---|---|---|---|---|---|
| | | Type | GR | SR | Type | GR | SR | |
| GPT-4o | ZS | 63.1 | 30.9 | 21.2 | 37.5 | 14.2 | 5.4 | 28.7 |
| Qwen2.5-VL-3B | ZS | 47.8 | 46.5 | 38.9 | 37.4 | 26.5 | 26.7 | 37.3 |
| Qwen2.5-VL-7B | ZS | 68.7 | 59.7 | 47.1 | 55.6 | 37.8 | 34.4 | 50.6 |
| OS-Atlas-4B | SFT | 49.0 | 49.5 | 22.8 | 49.6 | 34.6 | 20.3 | 37.6 |
| OS-Atlas-7B | SFT | 57.4 | 54.9 | 29.8 | 60.4 | 39.7 | 27.0 | 44.8 |
| NaviMaster-7B | SFT | **72.9** | - | 54.0 | 64.4 | - | 36.9 | - |
| UI-R1-3B | RFT | 57.8 | 55.7 | 45.4 | 52.2 | 34.5 | 32.5 | 46.4 |
| GUI-R1-3B | RFT | 58.0 | 56.2 | 45.4 | 52.2 | 34.5 | 32.5 | 46.5 |
| GUI-R1-7B | RFT | 71.6 | 65.6 | 51.7 | **65.5** | 43.6 | 38.8 | 56.1 |
| PAL-UI-3B | SFT | 60.4 | 58.7 | 49.3 | 54.8 | 36.9 | 34.6 | 47.8 |
| PAL-UI-7B | SFT | 71.3 | **70.5** | **57.8** | 65.1 | **46.8** | **41.7** | **58.9** |

## 5.2 Implementation Details

Following existing studies (Luo et al., 2025), we employ Qwen2.5-VL (Bai et al., 2025) as the backbone model and use Qwen2.5-VL-7B as the summary agent for its potential in basic GUI scene understanding. All training and evaluation processes are conducted on 8 NVIDIA A100-80G GPUs. For SFT training, we follow previous studies and utilize LLaMA-Factory framework (Zheng et al., 2024b). We set the global batch size to $8$ and learning rate to $1e-5$. During inference, we utilize the same prompt with unified tool-calling method across all experiments to ensure a fair comparison.

## 5.3 Results

Table 1 presents the comprehensive comparison with state-of-the-art methods in low-data setting, including zero-shot (ZS), supervised fine-tuning (SFT), and reinforcement fine-tuning (RFT).

**Comparison with Base Models.** Compared to the base models Qwen2.5-VL-3B and Qwen2.5-VL-7B under the zero-shot setting, our PAL-UI agents achieve substantial gains across all metrics. PAL-UI-7B reaches an overall score of 58.9%, an absolute improvement of 8.3% over Qwen2.5-VL-7B (50.6%). The boost is especially clear in Success Rate (SR): PAL-UI-7B achieves 57.8% on AndroidControl-High, surpassing the base model by 10.7%. Despite being trained only on Android-Control, PAL-UI also generalizes well to the out-of-domain GUI-Odyssey benchmark, where PAL-UI-7B improves SR to 41.7%, a 7.3% gain. These consistent improvements across both in-domain and out-of-domain tasks highlight the robustness and generalization capacity of our approach.

**Comparison with State-of-the-Art Methods.** PAL-UI further outperforms existing SFT and RFT baselines. PAL-UI-7B achieves the best overall score of 58.9%, surpassing GUI-R1-7B (56.1%) by 2.8% and outperforming the strongest SFT method NaviMaster-7B by a large margin. It establishes new state-of-the-art results in multiple metrics, including GR (70.5% on AndroidControl-High and 46.8% on GUI-Odyssey) and SR (57.8% and 41.7%, respectively). Notably, PAL-UI-7B outperforms GUI-R1-7B despite relying only on SFT rather than reinforcement learning, showing that our approach yields stronger results with lower training complexity. Finally, scaling from PAL-UI-3B to PAL-UI-7B consistently improves performance, validating the scalability of our method.

## 6 Further Analysis

**Ablation Study.** We investigate the contributions of different components in the PAL-UI framework, and the results are summarized in Table 2. Several key findings emerge: First, the **dual-level summary agent (SA)** consistently improves performance under both zero-shot and supervised fine-tuning settings. For example, on the out-of-domain GUI-Odyssey benchmark, equipping the base

Table 2: Ablation results. "SA" indicates summary agents. "PAL" indicates planning with active look-back mechanism. "Full" indicates our full approach, where we train the model on our constructed dataset and leverage SA and PAL during inference.

| Setting | AC-High | | GUI-Odessey | |
|---|---|---|---|---|
| | Type | SR | Type | SR |
| Zero-Shot | 68.7 | 47.1 | 55.6 | 34.4 |
| + SA | 70.9 | 54.6 | 62.8 | 38.9 |
| + SFT | **73.4** | 53.2 | 56.3 | 36.4 |
| + SFT & SA | 72.9 | 56.3 | 59.4 | 38.4 |
| + SFT & PAL | 67.4 | 52.3 | 56.6 | 37.1 |
| + Full | 71.3 | **57.8** | **65.1** | **41.7** |

Table 3: Comparison of context length and performance for different paradigm. *None*: not using memory; *+A*: using only actions as memory; *+ 5O* and *+ AO*: using recent 5 screenshot or all historical screenshots as memory; *+SA*: using summary agent for memory processing; *+ PAL*: using the full approach.

| Setting | Len. | AC-High | | GUI-Odessey | |
|---|---|---|---|---|---|
| | | Type | SR | Type | SR |
| None | 4307.6 | 65.4 | 45.8 | 52.6 | 34.6 |
| + A | 4371.7 | 68.7 | 47.1 | 55.6 | 34.4 |
| + 5O | 12630.0 | 69.1 | 48.3 | 56.8 | 36.4 |
| + AO | 16383.8 | 67.8 | 45.5 | 54.6 | 34.9 |
| + SA | 4965.4 | 70.9 | 54.6 | 62.8 | 38.9 |
| + PAL | 7330.2 | **71.3** | **57.8** | **65.1** | **41.7** |

Table 4: Cross-platform results of PAL-UI on Multimodal-Mind2web. We report the model performance on three benchmark splits: cross-task, cross-website, and cross-domain.

| Model | Cross-Task | | | Cross-Website | | | Cross-Domain | | |
|---|---|---|---|---|---|---|---|---|---|
| | Op.F1 | Ele.Acc | SR | Op.F1 | Ele.Acc | SR | Op.F1 | Ele.Acc | SR |
| Qwen2.5-VL-3B | 53.8 | 26.5 | 25.9 | 50.3 | 25.3 | 23.4 | 53.5 | 30.4 | 28.3 |
| Qwen2.5-VL-7B | 61.3 | 33.1 | 32.3 | 57.1 | 31.7 | 29.8 | 61.0 | 36.8 | 34.4 |
| PAL-UI-3B | 54.5 | 27.9 | 27.6 | 54.6 | 25.9 | 25.1 | 57.1 | 33.3 | 31.9 |
| PAL-UI-7B | 68.7 | 36.0 | 35.0 | 69.2 | 36.4 | 35.2 | 69.6 | 39.6 | 37.9 |

model with SA boosts SR from 34.4 to 38.9, confirming its effectiveness in mitigating the information loss inherent in action-only memory. Second, while **SFT alone** yields clear in-domain gains (SR on AC-High improves from 47.1 to 53.2), its effect on generalization is limited, with only a minor increase on GUI-Odyssey (34.4 to 36.4). This highlights the difficulty of transferring knowledge without a richer memory mechanism. Third, we observe that without memory **planning with active look-back (PAL)** hampers PAL-UI's effectiveness: as its performance drops on AC-High (SR from 53.2 to 52.3). This indicates that without rich historical summaries, the agent struggles to judge the relevance of retrieved information to make correct actions. Finally, the **full PAL-UI** framework, integrating SFT, SA, and PAL, achieves the strongest results across both benchmarks (SR 57.8 on AC-High and 41.7 on GUI-Odyssey). These findings show that the synergy of summarization and active retrieval is essential for robust long-horizon planning and cross-domain generalization.

**Cross-Platform Performance.** Although PAL-UI is trained exclusively on mobile data and primarily evaluated on mobile benchmarks, we further examine its ability to generalize across platforms. Specifically, we test PAL-UI on Multimodal-Mind2Web (Deng et al., 2023), an web navigation benchmark, following the official evaluation protocol and reporting Operation F1 (Op.F1), Element Accuracy (Ele.Acc), and Step Success Rate (SR). Results are shown in Table 4. Despite being trained solely on mobile trajectories, PAL-UI achieves consistent improvements on web tasks, with average gains of +3.8 and +2.4 points for the 7B and 3B models, respectively. While these gains are smaller than those observed in mobile domain, they still demonstrate that PAL-UI effectively transfers its planning capability to a different platform. This ability to maintain performance across different environments highlights the robustness and generalization potential of our approach.

**Context Length Comparison.** We compare the trade-off between context length and performance across three paradigms: PAL-based reasoning, action-only memory, and full screenshot memory. For evaluation, we select 50 trajectories from the AndroidControl test set and measure the input token length under each setting (Table 3). The results show that the screenshot-based method consumes the largest number of tokens yet provides little benefit, as redundant visual information increases both computational cost and prediction errors. By contrast, the action-only method requires

Table 5: Results of PAL-UI with different summarization models.

| Summarization Model | AC-High | | GUI-Odessey | |
|---|---|---|---|---|
| | Type | SR | Type | SR |
| Qwen2.5-VL-3B | 70.9 | 57.4 | 63.8 | 41.0 |
| Qwen2.5-VL-7B | 71.3 | 57.8 | 65.1 | 41.7 |
| InternVL3-2B | 69.4 | 56.2 | 64.2 | 39.8 |
| InternVL3-8B | 72.5 | 58.4 | 64.0 | 41.3 |

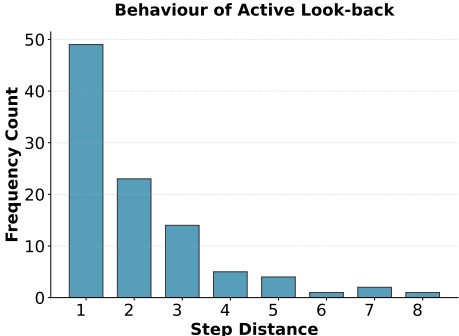

Figure 2: Behavior of active look-back.

the fewest tokens but yields clearly inferior performance due to the loss of critical historical details. Our PAL-UI strikes a balance between the two extremes: it adds only a modest number of tokens yet delivers significant improvements. Notably, PAL-UI achieves a 6% average performance gain while using just 44% of the tokens required by the screenshot-based approach, highlighting both the efficiency and effectiveness of our method.

**Effect of Summarization Models.** We conduct extensive experiments to investigate the impact of different summarization models on agent performance. Specifically, we leverage Qwen2.5-VL (Bai et al., 2025) and InternVL-3 (Zhu et al., 2025) series models for memory summarization. The results are presented in Table 5. As we observe, different summarization models do not exert a significant impact on overall performance. The reason might be that current MLLMs already possess strong OCR and visual comprehension capabilities, enabling them to accurately identify and extract key information in GUI screenshots and discerning action intentions and execution states across consecutive screenshots. As a result, we can employ a relatively compact model for memory compression to enhance agent performance while introducing minimal time overhead.

**Analysis of Retrieval Behavior.** We further analyze how the agent employs the retrieve tool during planning. Concretely, we sample 100 agent responses from the AndroidControl test set in which the retrieval tool is invoked, and record the distance between the retrieved step and the current step (Table 2). The results show that most retrievals occur within five steps of the current step, with the maximum distance extending to eight steps. This tendency likely reflects the relative simplicity of current navigation benchmarks: tasks usually involve short, localized interactions (e.g., simple GUI manipulations) that demand only limited long-range memory. We believe that in more complex or semantically demanding GUI environments, where long-distance dependencies play a larger role, the benefits of active look-back may become even more pronounced.

## 7 CONCLUSION

We presented PAL-UI (Planning with Active Look-back), a framework that empowers GUI agents to selectively retrieve past observations during planning rather than carrying the full history at every step. By combining dual-level summaries with an active retrieval tool, PAL-UI effectively balances efficiency and completeness, enabling agents to handle long-horizon tasks with improved accuracy and reduced context overhead. To support this paradigm, we introduced a deliberated look-back framework for constructing tool-augmented trajectories, yielding 8.6K high-quality instruction-tuning samples. Extensive experiments demonstrate that PAL-UI significantly outperforms both base MLLMs and state-of-the-art baselines on mobile navigation benchmarks, while also generalizing well to out-of-domain web environments. These results underscore the importance of active memory retrieval for robust GUI planning. Future work will explore extending PAL-UI to more complex tasks and environments, integrating reinforcement learning objectives, and broadening its applicability to real-world interactive systems.

# 8 ETHICS STATEMENT

In the course of this research, all procedures were conducted in strict compliance with established academic norms and ethical principles. The experimental data utilized fully adhere to ethical requirements, containing no personal private information, no material inconsistent with human values, and no biased or offensive content. The objective of this work is to enhance the capabilities of autonomous agents, with the ultimate aim of advancing AI technologies that can effectively benefit all of humanity and contribute positively to society and human welfare. In the writing process, LLMs were employed solely for the purpose of checking and correcting grammatical errors in the manuscript. All AI-generated content has been carefully reviewed by the authors to ensure the accuracy and rigor of the paper.

# 9 REPRODUCIBILITY STATEMENT

To ensure the reproducibility of our work, we provide a detailed description of our approach in Section 4. Moreover, we present the details about the implementation of our experiments in Section 5, including the detailed training and evaluation setting. Furthermore, we provide the code for evaluation in our supplementary material.

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
