# OpenReview forum: "PAL-UI: Planning with Active Look-back for Vision-Based GUI Agents"
_ICLR.cc/2026/Conference — ICLR 2026 Conference Withdrawn Submission_

### Official Review · Reviewer_EfSV · 2025-10-26

**Soundness:** 3
**Presentation:** 2
**Contribution:** 2
**Rating:** 4
**Confidence:** 3

**Summary:**

This paper introduces PAL-UI (Planning with Active Look-back), a novel framework that enhances the long-horizon reasoning ability of multimodal large language model (MLLM)-based GUI agents. Traditional GUI agents struggle with long-term dependencies due to limited memory, often losing essential visual context when truncating or textually summarizing history. PAL-UI addresses this by combining dual-level summarization—capturing both observation- and action-level information—with an active retrieval tool that enables selective recall of past screenshots when needed for planning.

**Strengths:**

- This work successfully pinpoints a challenging and essential perspective within the context of gui-agent.
- This work proposes modules such as summary and retrieval to address several issues in gui-agent tasks.

**Weaknesses:**

- The training method employed in this work is relatively simple, relying primarily on SFT without exploring advanced approaches.
- This work lacks a case study on the visualization of the entire pipeline.

**Questions:**

- While the article mentions in Lines 368–370 that PAL-UI adopts SFT due to its lower computational complexity compared to RL, performance should be prioritized. Computational complexity advantages should only be compared when RL’s performance is on par with that of SFT. As a method validated effective by numerous works, RL’s actual performance deserves to be demonstrated by the authors.
- I recommend that the authors demonstrate the cases of PAL-UI to provide readers with an intuitive performance evaluation.

---

### Official Review · Reviewer_jfZe · 2025-11-01

**Soundness:** 2
**Presentation:** 2
**Contribution:** 2
**Rating:** 4
**Confidence:** 3

**Summary:**

This paper proposes PAL-UI, a framework that enables GUI agents to actively retrieve past observations during long-horizon planning tasks. PAL-UI combines dual-level summarization with a retrieval tool that allows agents to recall specific historical screenshots on demand. The authors curate an 8.6K sample dataset, and train PAL-UI-3B and PAL-UI-7B models based on Qwen2.5-VL. Experiments on AndroidControl-High and GUI-Odyssey show PAL-UI achieves sota results.

**Strengths:**

- The paper focuses on an important research problem of GUI agents.
- The paper presents comprehensive experimental results and analysis.

**Weaknesses:**

- Seems the framework only combines a memory system with a retrieval tool. The framework design innovation seems limited.
- The paper provides no detailed failure mode analysis, which is important for agent research.
- Whether PAL-UI and baselines like GUI-R1 are trained on the same data.

**Questions:**

- Training data examples could be provided.
- Training and inference time comparison should be provided as well.

---

### Official Review · Reviewer_JZCh · 2025-11-02

**Soundness:** 3
**Presentation:** 3
**Contribution:** 3
**Rating:** 4
**Confidence:** 4

**Summary:**

This paper tackles a problem with GUI agents - they can't handle long tasks because visual history eats up too much memory. Instead of keeping all past screenshots, PAL-UI compresses history into text summaries. But it also gives the agent a "retrieval tool" to look back at specific screenshots when needed.

Evaluation results on mobile navigation tasks. It beats existing methods while using way less memory. The cool part is it transfers to web navigation without extra training. The agent mostly looks back just a few steps, which makes sense for current benchmarks.

**Strengths:**

PAL-UI introduces a smart "active look-back" mechanism that changes how GUI agents handle memory. Instead of cramming all visual history into context or just keeping recent stuff, it compresses everything into text summaries but lets the agent retrieve specific past screenshots on demand.

The method solves the scalability problem. Visual tokens are expensive and current approaches either lose important info or blow up context length. PAL-UI uses only 44% of the tokens compared to keeping all screenshots while getting 6% better performance.

The data construction process is pretty useful. Since regular demo data doesn't have retrieval examples, they created a four-stage framework to teach a teacher model when to look back.

**Weaknesses:**

The paper lacks detailed discussion of how the observation-level captioner and action-level validator actually work. What specific visual cues might get lost during text summarization?

The action-level validator also needs more examination. How does it handle cases where an action looks successful but actually fails? The paper should explore different summarization strategies. It would help to see quantitative analysis of what information gets retained versus lost.

The paper doesn't analyze edge cases where important UI elements could be missed. Things like subtle state changes, error messages, or visual feedback might not translate well to text.

It would be interesting to know how the agent figures out which specific step to grab without having to search through everything. Maybe there are ways to make this smarter, like using semantic similarity to find relevant past screenshots or allowing the agent to retrieve multiple related steps when needed.

The four-stage training process is neat, but it would be good to know more about how well it actually works. Are these four stages really the best way to teach retrieval behavior, or might other approaches work just as well?

**Questions:**

Please see my weakness section above for more detailed comments.

---

### Official Review · Reviewer_i2Sf · 2025-11-05

**Soundness:** 2
**Presentation:** 3
**Contribution:** 2
**Rating:** 4
**Confidence:** 4

**Summary:**

This paper tackles the challenge of memory limitations in GUI agents, which often fail long-horizon tasks because they cannot retain a full history of visual observations. The authors propose PAL-UI, a framework that reformulates the memory problem. Instead of naively storing all past screenshots, PAL-UI maintains a concise textual summary of the history and equips the agent with a retrieve tool. The tool allows the agent to look back and fetch a specific past screenshot on demand when it detects uncertainty. To train this behavior, the authors developed a deliberated look-back pipeline to curate a new 8.6K-sample dataset teaching the model when to use this tool. Experiments show that the resulting PAL-UI models significantly outperform baselines on complex mobile navigation tasks and, demonstrate zero-shot generalization to web navigation.

**Strengths:**

- The "active look-back" mechanism is a highly practical and clever solution to the context-length problem, balancing the efficiency of text summaries with the high-fidelity detail of visual history. It has benefits in certain scenarios compared to full history and more robust than text-only summaries.
- High-Quality Data Curation Pipeline. A significant contribution is the four-stage deliberated look-back framework used to create the training data. The data serves for teaching the agent when to be uncertain and use the tool.
- The proposed mechanism, as per the ablation study, is better than full context or purely text-based methods.

**Weaknesses:**

- Lack of evaluation on online envs, e.g., AndroidWorld or AndroidLab. The effectiveness of the look-back mechanism remains uncertain in real, dynamic environments.
- It is questionable that why the retrieve tool is designed as only receiving the index, rather than a typical RAG implementation based on semantics. It would be valuable to see if the model is sensitive to different design of retrieval.

**Questions:**

Please see weaknesses.

---

### Note · Authors · 2025-11-13

I have read and agree with the venue's withdrawal policy on behalf of myself and my co-authors.